# Seeing structural evolution of organic molecular nano-crystallites using 4D scanning confocal electron diffraction (4D-SCED)

Mingjian Wu [1✉], Christina Harreiß[1], Colin Ophus [2], Manuel Johnson[3], Rainer H. Fink [3] & Erdmann Spiecker [1✉]

Direct observation of organic molecular nanocrystals and their evolution using electron microscopy is extremely challenging, due to their radiation sensitivity and complex structure. Here, we introduce 4D-scanning confocal electron diffraction (4D-SCED), which enables direct in situ observation of bulk heterojunction (BHJ) thin films. 4D-SCED combines confocal electron optic setup with a pixelated detector to record focused spot-like diffraction patterns with high angular resolution, using an order of magnitude lower dose than previous methods. We apply it to study an active layer in organic solar cells, namely DRCN5T:PC$_{71}$BM BHJ thin films. Structural details of DRCN5T nano-crystallites oriented both in- and out-of-plane are imaged at ~5 nm resolution and dose budget of ~5 e$^-$/Å$^2$. We use in situ annealing to observe the growth of the donor crystals, evolution of the crystal orientation, and progressive enrichment of PC$_{71}$BM at interfaces. This highly dose-efficient method opens more possibilities for studying beam sensitive soft materials.

[1] Institute of Micro- and Nanostructure Research & Center for Nanoanalysis and Electron Microscopy (CENEM), Department of Materials Science, Friedrich-Alexander-Universität Erlangen-Nürnberg, Cauerstraße 3, D-91058 Erlangen, Germany. [2] National Center for Electron Microscopy, Molecular Foundry, Lawrence Berkeley National Laboratory, 1 Cyclotron Road, Berkeley, CA, USA. [3] Department of Chemistry and Pharmacy, Friedrich-Alexander-Universität Erlangen-Nürnberg, Egerlandstr. 3, 91058 Erlangen, Germany. ✉email: mingjian.wu@fau.de; erdmann.spiecker@fau.de

The properties of organic semiconductors and device performance, particularly in bulk heterojunction (BHJ) organic solar cells, are dictated by the nano-crystalline structure and morphology. This is due to the high anisotropy of opto-electronic properties of the constituent molecules or polymers, and their directional assembly into (semi-)crystals. The orientation relationship between molecule, crystal and morphological features, interface character of donor/acceptor components, degree of network percolation of the nano-scaled carrier transport channels in BHJ therefore govern the device performance, which all evolve sensitively depending on the processing conditions[1–5]. However, revealing the nanoscale structures at high spatial resolution using electron microscopy methods is challenged by radiation sensitivity of these soft materials[6,7] and the complexity of their structures. Diffraction imaging, also called four dimensional-scanning transmission electron microscopy (4D-STEM)[8], or nano-beam diffraction (NBD), with a small convergence angle $\alpha$ has recently demonstrated its power to reveal a multitude of nanoscale structural details in a very broad range of material samples, e.g. in refs. [9–11]. Mapping the orientation of $\pi$-stacking in organic semiconductor molecular crystals was recently demonstrated under cryogenic temperatures[12], opening a new application field in beam sensitive soft materials[13]. Cryo-freezing the samples is one of the general strategies to slow down the structural damage and extend the dose tolerance by up to an order of magnitude before the structures break down by the incident electron beam[7,12]. Working at cryogenic temperature, however, makes in situ observation of thermal induced structural evolution more difficult.

In 4D-STEM, pixelated detectors/cameras are used to record the full 2D diffraction pattern at each probed sample position, allowing full reciprocal space details of the scattered intensities to be analyzed afterwards. Radical developments have pushed the limits of detection efficiency and camera speed[13], as well as ever growing computational and software algorithms. The standard NBD setup available on most TEMs, is however, neither optimized for dose efficiency nor for angular resolution. With the term "angular resolution" we refer to the accuracy of locating Bragg diffraction disks/spots, rather than the angular sampling/resolution of electron scattering signals in general. A focused probe interacting with a small sample region results in high dose under a given probe current and far-field diffraction disks spread the signal over many pixels of the detector/camera (Fig. 1a). In applications of diffraction imaging, it is the position of Bragg reflections, their summed intensities, and their in-plane orientations which provide rich information on the crystalline phase[9], strain state[10,11] and orientation[12] of the underlying diffracting lattices. Although the distribution of intensity in the beam disks contain rich structural information, e.g., used in convergent beam electron diffraction[14–17] and ptychography[8,18,19], it is not useful for the aforementioned purposes but rather spreads the already low signals to many detector pixels. This lowers the SNR for a given detector, complicates the post-acquisition processing[13], and reduces the angular resolution because of disk overlap, when studying large unit cell samples such as organic crystals (typically few nanometers). For example, the Bragg angle for a 2 nm lattice spacing is only 0.6 mrad using 200 keV incident electrons, comparable to the probe semi-angle in many NBD experiments. Furthermore, at small diffraction angles, inelastic scattering contributes a strong background (cf. Fig. S1), reducing the diffraction signal-to-background ratio (SBR). Applying low doses, diffraction disks become faint or intensities within disks become even sparse and locating their position on top of the inelastic background will be very challenging even with state-of-the-art direct detection cameras with DQE approaching unity.

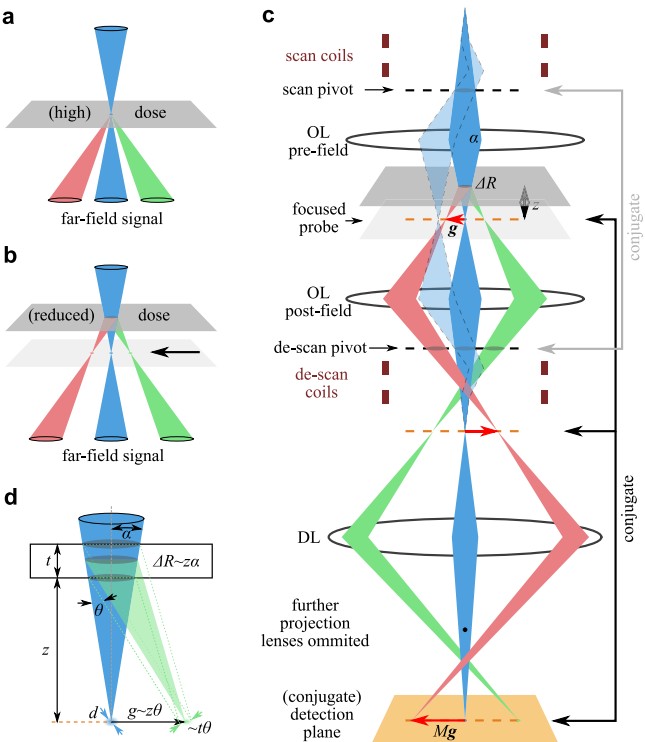

**Fig. 1 The 4D-SCED setup. a** Typical STEM setup where detector(s) is(are) located at far field. **b** Defocusing the probe mitigates limited dose budget for radiative sensitive samples. **c** Scheme of a simplified optic path to realize scanning confocal electron diffraction. OL objective lens, DL diffraction lens. **d** A simple geometric consideration of spatial and angular resolution.

To mitigate the challenge of limited dose budget, we can defocus the probe, and/or use a shorter camera length, i.e. apply angular under-sampling to enhance SNR. However, both approaches would not alleviate the problem of diffraction disk overlapping and poor SBR at low angles. Due to coupling between the diffraction signal disks and illumination convergence in a STEM setup, reducing the convergence angle is sought to be a natural solution. This could be achieved by using (1) condenser zoom, (2) customized small probe-defining apertures as recently explored[13], or (3) an objective lens with longer focal length, e.g. condenser mini lens in low-mag STEM or Lorentz STEM[20]. Condenser zoom can lower the convergence semi-angle to 0.5–1 mrad[21], which can still cause disk overlap for lattice planes of ~1.2–2.4 nm distances using a 200 keV primary beam. Customized apertures require replacement of the standard apertures, reducing the flexibility of the instrument. In addition, cutting down the convergence angle with a smaller aperture by a factor of $n$ lead to lowering of current by a factor of $n^2$ which can result in a very low beam current and may make the experiments challenging to perform. Objective lenses with long focal length suffer from very large aberrations[20], as probe correction of these weak lenses are not routinely available. Furthermore, using a weak objective lens means a long camera length, causing diffraction patterns fall beyond the camera/detector field of view. Dedicated alignment of the projection system may shrink the effective camera length. However, medium to high angle scattering would still be blocked by the differential pump aperture in the projection chamber. Therefore, for diffraction imaging studies of soft materials, we want to develop alternative flexible methods, optimized for both dose efficiency and angular resolution, while retaining sufficient spatial resolution.

Here, we introduce a diffraction imaging modality which, in contrast to 4D-STEM, uses the imaging mode of STEM rather than the diffraction mode. This confocal optical setting combines small convergence angles $\alpha$ with a large sample defocus $z$ (i.e., a pencil beam illumination) to obtain sharp diffraction spots at the confocal plane. Similar to earlier endeavor of low-dose methods[22], this is optimized for beam sensitive samples, which is different from typical scanning confocal electron microscopy (SCEM) with in focus illumination and large convergence angles for achieving ultra-high lateral and depth resolution[23–27]. In this mode, the spatial and angular resolution can be tuned to adapt to the dose budget and largely decoupled from convergence of illumination. We call this technique 4D scanning confocal electron diffraction (4D-SCED) and use it to study the structure of organic semiconductor thin films and molecular nano-crystallites in BHJ. We show that the 4D-SCED method (1) has high angular resolution for investigating the rich structural information, and (2) can reduce dose by about an order of magnitude compared to the standard NBD setup used in many 4D-STEM applications. We further demonstrate that 4D-SCED even enables in situ monitoring of structural evolution and growth of nano-crystallites in BHJ at elevated temperatures.

## Results

**The 4D-SCED setup.** In a diffraction-limited STEM setup, where small convergence angles are applied, and lens aberration can be ignored, spatial and angular resolution is well described by the Abbe equation:

$$d \approx \frac{\lambda}{2 \sin \alpha}. \qquad (1)$$

In a typical NBD setup, the sample plane is coincident with the plane of the probe with sharply focused probe of size $d$ at the left side of the equation; and detector is at far field, or conjugate plane of the aperture, which defines $\alpha$, thus disk patterns are detected (Fig. 1a). Due to the reciprocal relationship, one can optimize spatial or angular resolution on either side of the equation for a given incident electron wavelength. In view of the reciprocal relationship, exchanging the planes of probing and detection would allow for an extended illumination area (reduced dose) and sharply focused diffraction spots (high SNR and angular resolution) without altering apertures.

Examining the beam path in the case of a defocused probe (obtained by raising the sample, Fig. 1b), the cross-over of the direct and Bragg-diffracted beams are generated between the sample and the far-field diffraction pattern. Detection of these cross-over points can be easily realized using a confocal optics setting (Fig. 1c). In this setup, the detector plane is set to the confocal plane which is conjugate to the cross-over pattern (focal plane), i.e., the diffraction/intermediate lens is working in imaging mode. Focused diffraction information is then formed at the confocal plane. Lowering the sample relative to the beam focus also results in a diffraction spot pattern at the confocal plane, but the pattern is rotated by 180°. This optical setting is similar to that used in large-angle convergent beam electron diffraction (LACBED) in TEM mode[28,29], as well similar TEM methods to obtain spot diffraction patterns (without scanning, and thus not spatially resolved)[30,31]. In these cases, spatial resolution was not of major concern, and a large convergence angle (up to a few degrees) and large defocus (tens of $\mu$m to mm) were applied. Scanning (shifting) the probe over the sample creates a 2D real space image grid, therefore de-scan of the probe after the imaging lens is required to stabilize the image of probe (and thus also the diffraction patterns) on the detection plane (Fig. 1c). This requirement is typical for SCEM. Since it is the diffraction information which is of interest, and the full

diffraction pattern is recorded in 4D datasets, we call this setup 4D scanning confocal electron diffraction. To balance spatial and angular resolution 4D-SCED uses a small convergence angle and appropriate defocus value, as will be discussed in the following.

**Resolution of 4D-SCED: geometric considerations.** We use a simple geometric model (Fig. 1d) to examine the experimental parameters and discuss the achievable spatial and angular resolution. As illustrated, the diffraction "spots" in SCED are not identical to that of real far-field pattern, e.g., selected area electron diffraction (SAED) using parallel illumination, due to the $z$-sensitivity of scattering signals in the confocal setup[32]. The separation of the diffraction information, $g \approx z\theta$ ($\theta$ is twice the Bragg angle), is dependent on the defocus $z$, while the spread of the diffraction spots depends on (i) the size of the Airy disc at cross-over (see below) and (ii) the sample thickness $t$. The latter can be estimated geometrically to be of the order of $t\theta$. At defocus values comparable to sample thickness, the complete wave-optical simulation and full dynamical beam-specimen interaction should be considered, which has been explored in detail with the goal of extracting high-resolution 3D information in SCEM, e.g. in refs. [27,33]. To obtain sharp spot patterns for nanoscale crystallography studies, we want to suppress the spread of the diffraction spots ($\sim t\theta$) to be far smaller than the separation ($\sim z\theta$), giving the condition $z \gg t$. In addition, a homogeneously thin, flat sample without tilt/bending is desirable in order to position a region of interest at the same defocus $z$. For a typical TEM specimen with $t < 100$ nm, a defocus $z > 2\,\mu$m already results in a sharp spot diffraction pattern (cf. Fig. S2). For a given defocus, the spatial resolution is governed by the interaction area of the probe which is determined approximately by the convergence semi-angle $\alpha$ and defocus $z$ via (Fig. 1d)

$$\Delta R \approx z\alpha \qquad (2)$$

Therefore, at a given defocus $z$, a smaller convergence angle $\alpha$ is preferred to gain higher spatial resolution. For small probe convergence, the probe size $d$ at focus is limited by diffraction via the Abbe relationship given in Eq. (1). The angular resolution can then be estimated to be on the order of $d/z$[30]. Nevertheless, due to the reciprocal relation between spatial and angular resolution governed by the Abbe equation (1), one has to trade-off spatial and angular resolution also in an SCED setup, e.g., increasing $\alpha$ improves the angular resolution through reducing the in-focus spot size, but Eq. (2) shows that increasing $\alpha$ worsens the spatial resolution, i.e., $\Delta R$ become larger. A similar trade-off also applies to defocus $z$. Larger $z$ gives better angular resolution but worsened spatial resolution. These parameters have to be balanced and can be tuned to the needs of the experiment. For a 200 keV primary beam energy ($\lambda = 2.504$ pm) and a convergence semi-angle of $\alpha = 1$ mrad, a probe size of ~1.2 nm can be achieved. With defocus set to $z = 5\,\mu$m, a spatial resolution of ~5 nm and angular resolution of ~0.24 mrad would be expected.

Actually, due to the delocalization of electron beam damage in soft materials[34], and the long tail of the Airy fringes in a focused probe, spatially under-sampling is usually needed[13], thus one will not sacrifice achievable spatial resolution too much in dose-limited cases. Furthermore, defocusing the probe result in a more homogeneous illumination on a larger sample region thus a higher ratio of signal to dose is expected compared to in-focus illumination at identical spatial sampling and beam flux. Finally, the chromatic aberration of the post-specimen lenses will cause a chromatical defocus spread of the beam spots. The chromatic defocus of inelastically scattered electrons with energy loss of carbon $K$-edge ($\Delta E = 285$ eV) is $\Delta f_C = C_c \Delta E/E_0 \approx 3.5\,\mu$m for $E_0 = 200$ keV and typical chromatic aberration coefficient

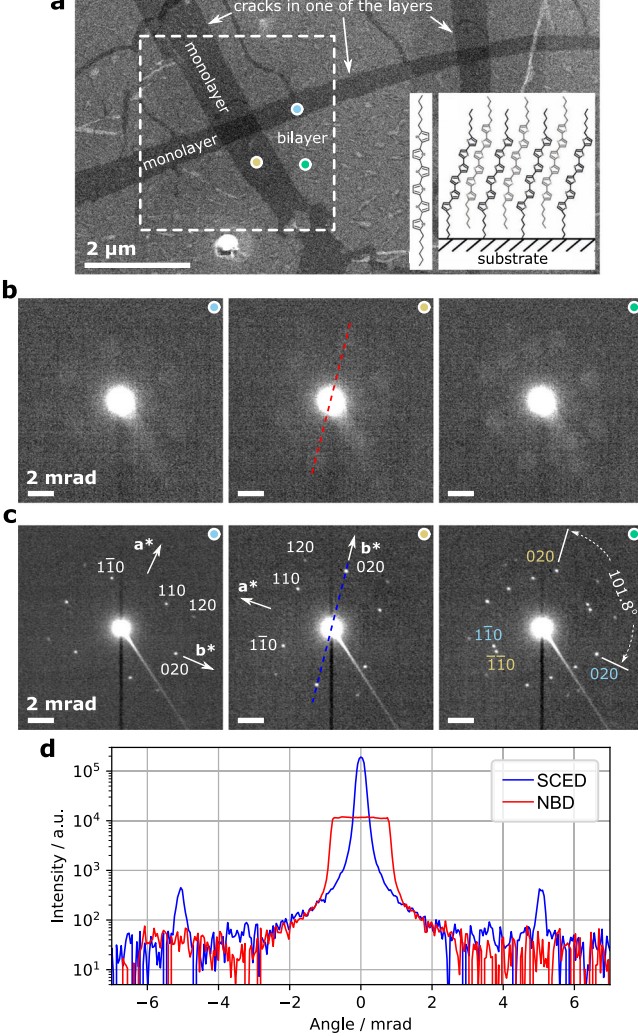

**Fig. 2 Comparison of NBD and SCED results of a 2D single crystal thin film of $\alpha, \omega$-DH6T bilayer. a** STEM-ADF image of the area of interest (acquired at the end of experiment). Insets show the molecular structure and scheme of the assembled monolayer on the substrate. The white box indicates where 4D-NBD and 4D-SCED data were acquired subsequently. Three representative raw patterns extracted from the positions marked with color dots are shown in **b** for the NBD results and **c** SCED results. **d** intensity profiles of the raw pattern comparing NBD and SCED.

sample, spatial resolution is not so important, e.g., for revealing crystal orientation or rotation angles between layers, thus we applied a large defocus of $z = 30\,\mu m$ (~50 nm illumination diameter) to reduce dose and allow acquiring both 4D-NBD and 4D-SCED datasets subsequently from same region for comparison. The sample was prepared via self-assembly on liquid surface[37] and transferred to TEM grid for methodology study here. Cracks formed during drying of the thin film allowing crystal orientation at regions of mono- and bilayer to be studied (Fig. 2a). The critical dose using 300 keV incident electrons as evaluated based on fading of diffraction spots is in the range of 20–50 e⁻/Å², depending on the number of layers. The white box in Fig. 2a show the area where datasets were acquired using identical dose of ~1.3 e⁻/Å². 4D-SCED was acquired after the 4D-NBD experiment where the structure was partly damaged. Comparing the raw diffraction patterns (Fig. 2b, c), an increased diffraction signal peak by about an order-of-magnitude is obviously seen. The focused diffraction spots in SCED result in high angular resolution thus the {110} diffraction spots from the two respective layers are clearly separated without overlap, while the corresponding discs are hardly discernible in NBD. The in-plane rotation of 101.8° of the two stacked layers can be easily determined from a single frame SCED pattern without sophisticated noise reduction and disk detection[38] typically required for a similar analysis based on NBD datasets. We emphasize here that the smallest available convergence of 0.7 mrad on our instrument was applied (cf. section methods). With customized smaller apertures, thus smaller beam disks, more concentrated diffraction signals are expected, and the difference would be less striking.

**Characterize bulk heterojunction using 4D-SCED.** With 4D-SCED, we further characterize an active layer of BHJ, and evaluate its figure of merit on such structurally complex soft material, where high spatial resolution is required to reveal the nanocrystalline domains. For this purpose, we choose a BHJ film comprising a blend of a small molecule thiophene-based electron donor DRCN5T and a fullerene acceptor $PC_{71}BM$ which has been treated by solvent vapor annealing (SVA) in $CS_2$ for 840 s. Solar cells based on this system have been reported to show high efficiency and good stability[3]. Thermal annealing (TA) and SVA are typical strategies in processing BHJs to tune their morphology via phase decomposition and coarsening and optimize the device performance[39]. The nanomorphology and crystallinity of this material system present a clear correlation to device performance, which has been shown to depend sensitively on the processing conditions[2]. In literature, nanomorphology of BHJs are routinely studied using atomic force microscopy (AFM) topology maps or bright field TEM images, although these methods may render ambiguous picture due to the lack of direct chemical and/or structural contrast. Spatially resolved analytical methods, including energy dispersive X-ray spectroscopy (EDXS), electron energy loss spectroscopy (EELS) and energy filtered TEM (EFTEM), provide elemental specific signals, which can be used to reveal the nanomorphology of binary or even ternary BHJs unambiguously in systems with sufficient chemical contrast. For example, thiophene-based donors (sulfur-rich) and fullerene-based acceptors (carbon-rich) can be well unraveled using carbon and sulfur signals, respectively[3,40,41] (see also Figs. S5–S7). However, these analytical methods do not provide structural/crystalline information and require electron dosage orders-of-magnitude higher than the critical dose of structural damage[7]. In our recent structural analysis of this material system using grazing incidence wide-angle X-ray scattering (GIWAXS), the unit cell structure of the small molecule crystal DRCN5T was determined, and a coexistence of (molecular) face-on and edge-

$C_C = 2.5\,mm$[35], in the same order to the defocus applied in SCED. In SCED, we use on-axis illumination therefore the chromatically defocused disks are concentric to the diffraction spots[36], thus do not influence the accuracy to locate the Bragg spots. Nevertheless, this results in a reduced diffraction SBR, with the degree of reduction additionally depending on sample thickness. A more quantitative insight of all the aforementioned aspects requires wave-optical simulation, which will be addressed in a follow-up work. However, the ultimate achievable spatial resolution of beam-sensitive samples is typically determined by the available dose budget and detection efficiency, as well as speed of the used camera (cf. method section).

**Dose effectiveness of 4D-SCED.** We first demonstrate the figure of merit of 4D-SCED compared to 4D-NBD (using standard apertures) with a single crystal thin film of organic semiconductor $\alpha, \omega$-DH6T bilayer. For this high quality 2D single-crystalline

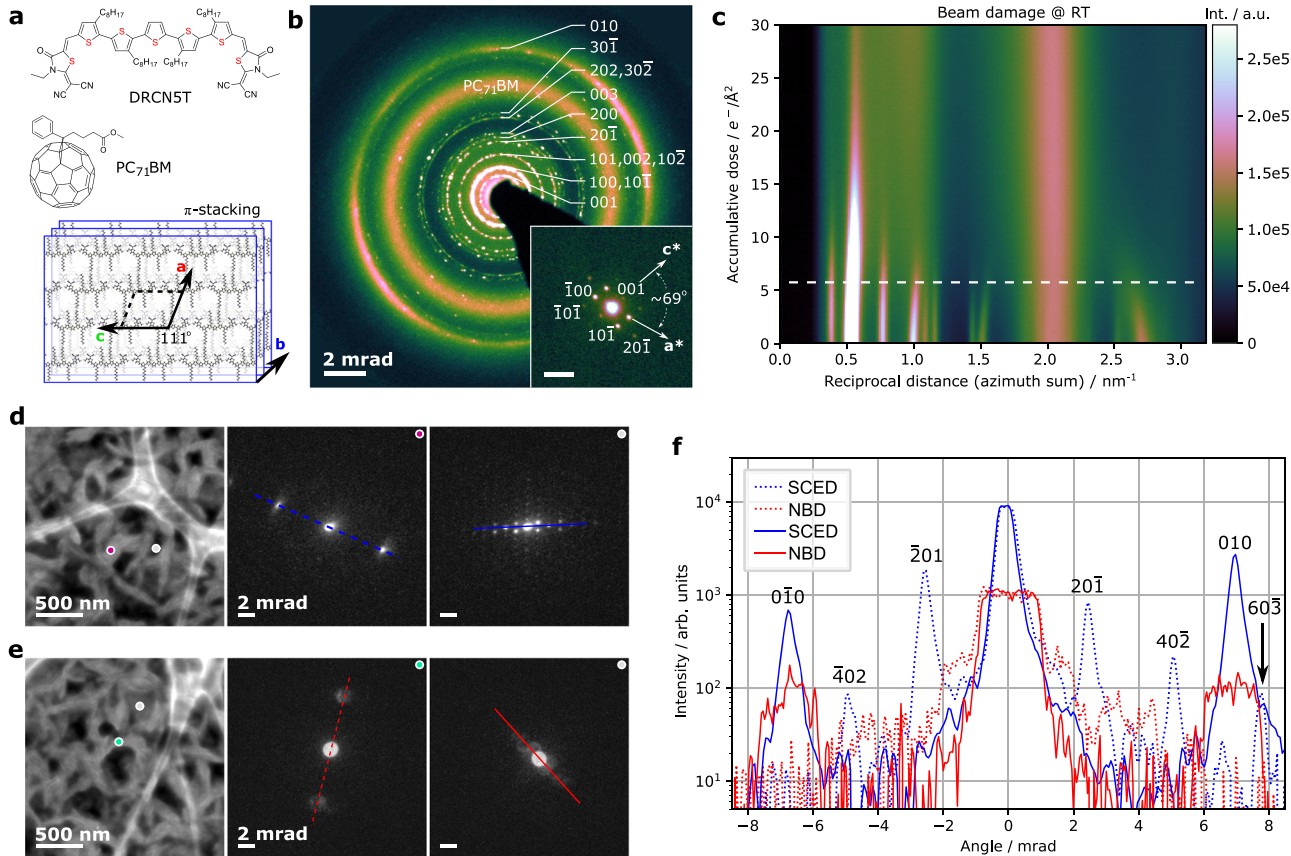

**Fig. 3 Comparison of raw patterns of SCED and NBD using a highly beam sensitive, structurally complex BHJ thin film of DRCN5T:PC71BM blend after SVA in CS2 for 840 s. a** molecular structure of donor component DRCN5T, acceptor component PC71BM and scheme of DRCN5T crystal. **b** (elastically filtered) SAED pattern from a sample region of 3.5 μm across. The inset (scale bar: 2 mrad) shows an indexed single diffraction pattern of a face-on DRCN5T crystalline domain extracted from a SCED dataset (without elastic filtering). While elastic filtering is necessary to enhance SBR (cf. Fig. S1), it is not needed in SCED. **c** azimuth integrated SAED as function of the accumulative dose at RT, results at cryo temperature shown in Fig. S3. **d** SCEM-ADF image simultaneously obtained during the SCED data acquisition. **e** STEM-ADF image simultaneously obtained during NBD data acquisition. Single raw diffraction patterns extracted from the marked dots in **d** and **e** representing the edge-on and face-on DRCN5T nano-crystallites respectively, are extracted and show on the right side. **f** Line profiles comparing the signals of raw patterns represent more than an order of magnitude higher SNR, signal analysis of the whole pattern is shown in Fig. S4.

on crystalline domains was deduced[42]. The molecular orientation, particularly the π-stacking orientation, within the nanoscale percolating network is the key to understand the transport properties. However, the important question of how the molecular orientation locally relates to the nanomorphology of the donor/acceptor blend and how both evolve upon processing have not yet been answered.

Figure 3a shows the molecular structure of PC71BM and DRCN5T, respectively. The in-plane tiling and out-of-plane stacking of the small molecule is also schematically depicted. Figure 3b presents an elastically filtered SAED (cf. method and Fig. S1) pattern of the sample acquired at room temperature (RT) with a total electron dose of ~0.8 e⁻/Å². In the experiment, the diffraction rings fade out very rapidly. A time series (and thus accumulated dose) of the SAED was recorded, and the azimuthally integrated profiles as function of accumulative dose is plotted in Fig. 3c. While the in-plane molecule planes, represented by the {100} diffraction ring at 0.55 nm$^{-1}$, survived beyond ~15 e⁻/Å², the π-stacking, corresponding to the {010} diffraction ring at 2.65 nm$^{-1}$, can only withstand a beam dose below ~5 e⁻/Å². We note that this critical dose is comparable to that of polyethylene and several times lower than that for P3HT in P3HT:PCBM, an extensively studied material system for organic solar cells, which shows a critical dose of 16–19 e⁻/Å²[43].

It is orders of magnitude more vulnerable than metal-organic frameworks, which can tolerate ~100–1000 e⁻/Å², as measured in controlled experiments[7]. The individual diffraction rings follow different trends upon beam bombardment that reveal the time evolution of structural damage. It is apparent that almost all in-plane diffraction rings {h0l} expand to higher values while the {010} diffraction ring shrinks, indicating that the crystal order of the π-stacking expands immediately upon beam bombardment while the in-plane packing of the molecule is gradually condensed, likely due to mass losses of the flexible side chains. Working under cryogenic temperature helps to preserve the crystalline order of the small molecule to about four times the total electron dose compared to RT (Fig. S3). However, ice formation was observed (in both SAED and real-space imaging) upon illumination. This disturbed the analysis of the native structure of the sample. All our experiments were therefore performed at RT.

To illustrate the enhanced SNR of 4D-SCED compared to NBD 4D-STEM using standard apertures, we compare the datasets acquired from the same sample at neighboring fresh areas, under identical dose conditions. We emphasize here that the smallest available convergence semi-angle of 0.85 mrad on our instrument was applied. Figure 3d–e show the sample areas and representative raw diffraction patterns extracted from the marked positions

using 4D-SCED (Fig. 3d) and NBD 4D-STEM (Fig. 3e), respectively. Each of the two patterns represents edge-on (color dot) and face-on crystallites (gray dot). In the face-on case the spots from the {h0l} planes, separated by angles < 1 mrad are clearly resolved in the SCED mode showing very high signal intensity, while the corresponding discs are heavily overlapping in the NBD mode. Apart from the disc overlap, the SNR is much inferior to that in SCED. At same beam flux, the difference of SNR will be dependent on the exact shape of probe profile and area of electron-sample interaction in the respective setups. In these comparisons (Figs. 2 and 3), illumination conditions, sample defocus and the dwell time were set to identical values. The difference in SNR is therefore solely from the imaging optics. Line intensity profiles from the NDB and SCED data are extracted along the blue and red lines, respectively, and are shown in Fig. 3c. The dashed lines compare data from the edge-on domains while solid lines compare the data from face-on domains. Obviously, almost all diffraction peaks in 4D-SCED show an order-of-magnitude higher intensity compared to NBD. This is most prominently revealed by the {603} peaks which are clearly visible in SCED, while in NDB they are buried in the noise floor. Comparing the SNR from the whole maps (cf. Fig. S4) reveals an average enhancement of the peak intensity by an order-of-magnitude in 4D-SCED. Since the noise level (~10 counts in current case) is intrinsic to the camera, the SNR level in SCED is therefore about an order-of-magnitude higher than NDB under the applied acquisition conditions. This means under same detectability criteria of our camera, SCED requires much less dose for Bragg peak detection, corresponding to significantly enhanced dose efficiency. Again, the difference would be reduced if smaller apertures would be available.

With its high angular resolution, the patterns acquired via SCED enables mapping of the orientation of nano-crystallites not only in edge-on (large diffraction angles) but also in face-on orientation (small diffraction angles). Furthermore, the sharp diffraction spots in SCED can be used to locally analyze the crystallographic structure of individual nano-crystallites, which is more difficult in the comparable NBD experiments. To demonstrate this, the inset in Fig. 3b shows an indexed SCED pattern extracted from a face-on domain, which agrees well to that obtained from GIWAXS studies[42]. Since the charge carrier transport properties of molecular crystals are highly anisotropic, the orientation of the molecular crystals in the domains and the micrometer scale percolation of the domains are critical to pinpoint the device performance. Figure 4 visualizes the orientation of edge-on crystal domains using the color wheel method, and the location of the face-on domains are displayed as superimposed grayscale maps. The short color segments represent the backbone of DRCN5T molecules, which is perpendicular to the {010} diffraction g-vectors (schematically shown in Fig. 4a). It is now clear that the DRCN5T molecular planes of the π-stacking are oriented parallel to the long axis of the nanoscale fiber structures, and the π-stacking direction, i.e., [010], along the short axis of fibers. In this sample (SVA treated in CS₂ for 840 s), the face-on domains cover roughly circular areas ranging from a few tens up to a several hundred nanometers in size. Within a face-on domain, the molecular crystals show tilted and twisted diffraction patterns (top-right insets in Fig. 4c) indicating either a strong bending of the domain or the existence of sub-grains and grain boundaries. Furthermore, successive raw diffraction patterns across the interface of edge-on domains reveal that PC₇₁BM is enriched mainly at the donor interface. This is best visualized in the (virtual) annual dark-field image using the diffraction information characteristic to PC₇₁BM, i.e., between 1.8 and 2.3 nm⁻¹, in Fig. 4d. We note that mapping the PC₇₁BM was not feasible in

our NBD 4D-STEM datasets using a simple virtual aperture method due to the much inferior angular resolution and overlapping diffraction information. In samples which have undergone SVA using CHCl₃ as the solvent, similar fiber-like edge-on domains (in projection) were observed but the face-on domains also resemble a fiber-like shape (cf. Fig. S6), which is different to the more circular shape after SVA in CS₂. Considering that the DRCN5T molecule has a flat π-conjugate plane and its flat tiling into crystalline grains, charge carrier hopping between π-stacking planes, i.e., along the molecular crystal [010] direction, is not likely[44]. The high-mobility direction of charge carrier transfer is dominated by pathways in the crystal a-c plane, which is determined to coincide with the long axes of the fiber.

**In situ observations**. Finally, we use 4D-SCED to study the structural evolution of DRCN5T:PC₇₁BM during TA by in situ heating the sample thin film in the vacuum of TEM. Inset in Fig. 5a schematically shows the temperature profile applied during the in situ experiment. 4D-SCED datasets were recorded sequentially with only few seconds between scans, to move the sample to a nearby fresh (un-illuminated) region. ADF image is also simultaneously recorded during 4D-SCED data acquisition. Data acquired directly after the temperature ramp suffer from heavy thermal drift (sample shift during data acquisition), and the images are therefore heavily distorted. Figures 5a–e show visualizations of the sample morphology and orientation of donor crystallites at different TA steps. Examining only the ADF images (dominated by mass-thickness contrast), the contrast seems to suggest an interconnected and continuous morphology up to annealing at 120 °C for ~8 min, similar to literature reports[3], and in accordance to the morphology revealed with EFTEM in our ex situ annealing experiment (cf. Fig. S7). However, the true structure is unraveled to be quite different using 4D-SCED which provides crystal information. The seemingly continuous donor phase is actually composed of small crystalline domains already in the as cast state. A certain degree of π-stacking ordering is visible, but barely any face-on domains. Very small (<30 nm long) edge-on domains are visible, showing a rod-like shape in projection. The acceptor component PC₇₁BM appears homogeneous. The horizontal streaking in these maps is more prominent in the in situ experiment suggesting that sample charging takes place, which may accelerate the structural damage and further lower the critical dose[6,7]. After the sample has annealed under 100 °C for ~8 min, a few tiny face-on domains spanning only 3–4 probed pixels (15–20 nm) start to emerge. Here, the growth of the edge-on domains in size is not yet obvious. After raising the temperature to 120 °C and holding for about 6.5 min, growth of both edge-on and face-on domains are observed, which is accompanied by phase separation of the PC₇₁BM. Further raising the temperature to 140 °C and holding it for 8 min, the edge-on domains become sharper and enrichment of PC₇₁BM towards the edge-on DRCN5T domains are observed. Finally, after the sample has been held at 160 °C for about 10 mins, the edge-on domains grow to an apparent length of 300–500 nm and large face-on domains also appear. Further annealing does not result in additional growth, which is likely due to the completed phase separation and depletion of the small molecules. Interestingly, during the growth of fiber-like edge-on crystalline domains, the aspect ratio of about 3–4 seems to remain throughout the entire thermal annealing processing. With the observed crystallographic orientation relationship to the fiber morphology, this indicates that the growth of the nano-crystallites must be faster in the crystallographic a-c plane, while the crystallites expand much slower in the π-stacking direction [010]. This further hints that the growth (attachment of

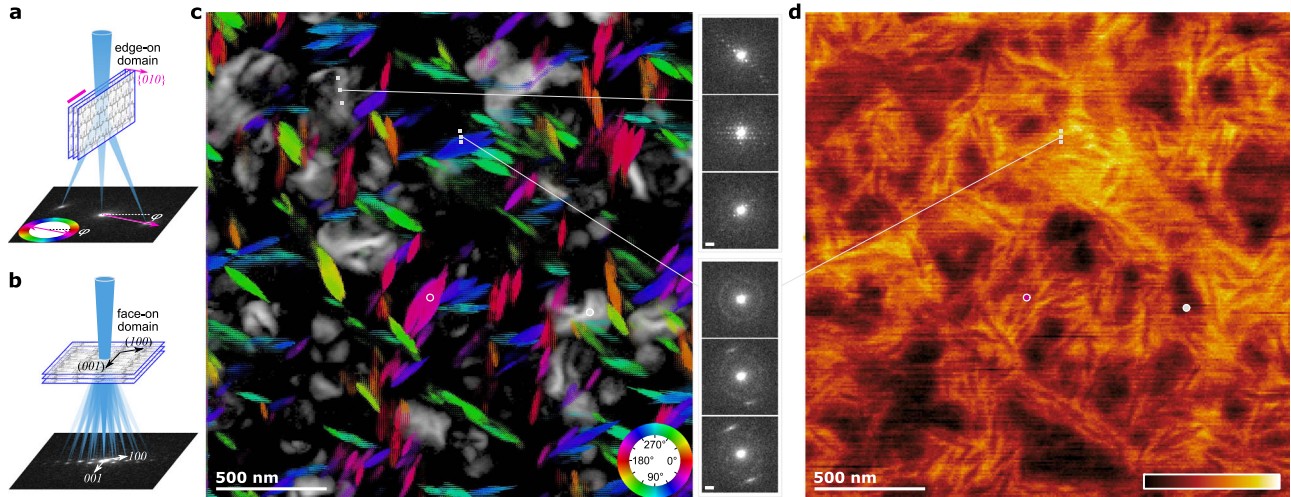

**Fig. 4 Visualization of the orientation of donor nano-crystallites and distribution of acceptor.** Scheme **a** uses a color wheel method to encode the edge-on domain orientation at a probed location; **b** demonstrates our ability to determine the grain orientation of the face-on domains. **c** Visualization of the whole scanned field of view. Insets on the right (scale bars: 2 mrad) are raw diffraction patterns extracted from the white dots. **d** Mapping of acceptor distribution using virtual annual aperture including only the angular range of $PC_{71}BM$ in each diffraction pattern (cf. Fig. 3b), the mapping is shown in "black body" color representation i.e., brighter means higher diffraction signals of $PC_{71}BM$.

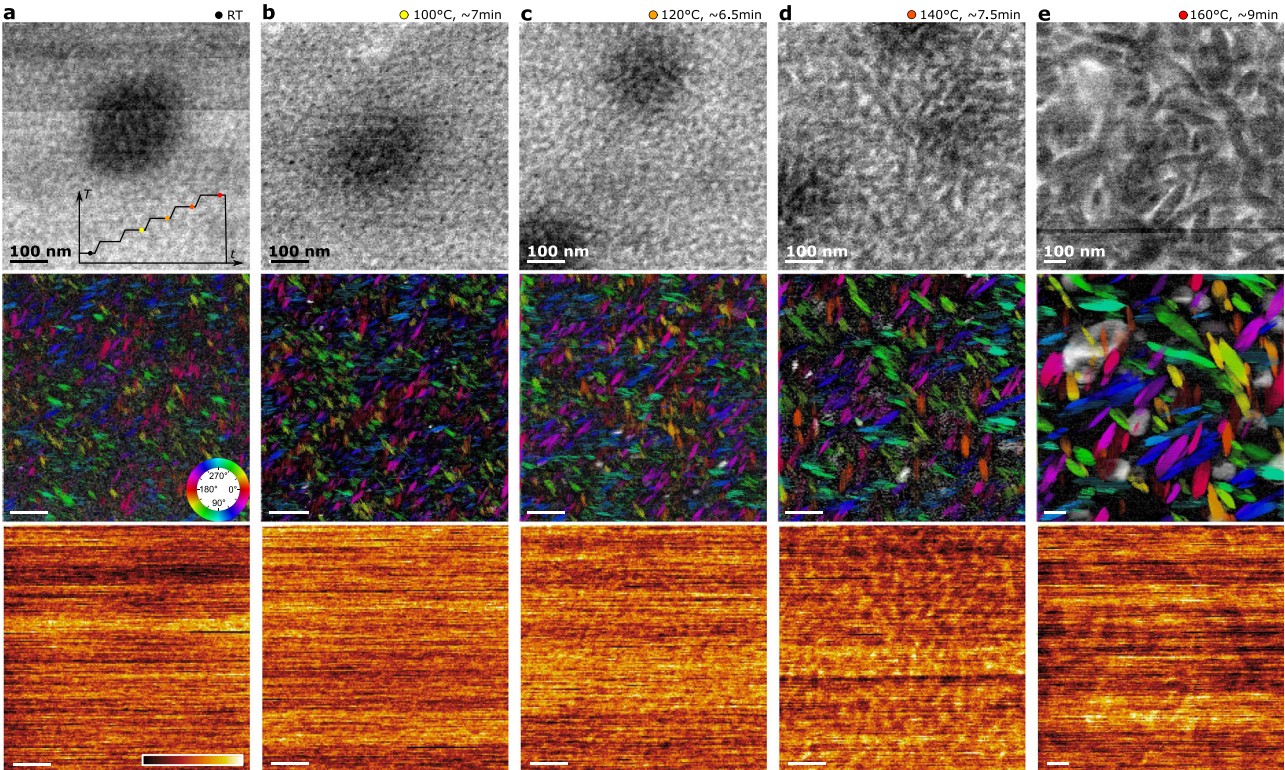

**Fig. 5 Observing the structural evolution upon thermal annealing of DRCN5T:PCBM thin film in the TEM.** A continuous "ramp-and-hold" temperature profile (inset in a) was applied. The data at each stage is visualized in columns (**a–e**), with the holding temperature and time indicated. The top raw show the simultaneously acquired ADF image; the middle and bottom rows are visualized 4D-SCED datasets using the same scheme as in Fig. 4. Scale bars: 100 nm. Note that the field of view in **e** was increased during experiment to increase the chance to capture face-on domains.

new molecules to crystalline seeds) via out-of-plane $\pi$-electron interaction is slower than the in-plane growth.

## Discussion

Successful and meaningful in situ observations of beam sensitive materials relays on many factors. The most important aspect is

the dose budge of the sample at relevant conditions. The electron beam induces permanent, irreversible impact (e.g., generation of charged states, radicals, etc.), even at vanishingly low dose[6,7,34], which influences the kinetics of any dynamic processes. In our experiment, we noticed that regions with illumination history were "dead", and no further growth or ripening would follow at later stage of temperature and time (also independently evaluated,

as shown in Fig. S8). This is likely the reason that no growth of the structures were observed in any single frame of 4D-SCED datasets (Fig. 5a–e) (remember that data acquisition from top to down took about 2 min). Setting the experimental conditions below critical dose, the beam freezes the kinetics upon first illumination, but still allow the structure to be probed before damage. Move to fresh sample regions is necessary to record the true structure evolution. To this end, quantitative knowledge gained from in situ observations are only meaningful if it can be validated via ex situ snapshots. The 4D-SCED setup optimizes the ratio of structure information to dose for such samples and the parameters should be chosen to accommodate the available dose budget and target spatial resolution. In the current work, scintillator coupled CMOS cameras were used. Applying state-of-the-art direct detection camera with DQE approaching 1 and higher frame rates is expected to further improve the spatial resolution (cf. methods section).

In conclusion, we have demonstrated 4D-SCED as a highly dose-efficient and high angular resolution diffraction imaging method. The benefit of improved information to dose ratio in this setup is mainly attributed to (1) the more homogeneous electron beam sample interaction obtained with (defocused) pencil beam illumination and (2) simultaneously guaranteed spot-like diffraction signals, boosting both SNR and SBR, which is largely decoupled from illumination convergence. Under optimized acquisition parameters, we observed the annealing-induced growth and structural evolution of nano-crystallites at sub-5-nm spatial resolution, under a dose budget of ~5 $e^-$/Å$^2$. When coupled to state-of-the-art high DQE detectors and more advanced data mining algorithms to extract subtle signals out of the large datasets like those used in NBD studies, we expect further improvements in the dose efficiency. We believe this 4D-STEM modality will open exciting possibilities in the study of nano-crystallography of soft materials.

## Methods

**Sample preparation**. The bulk heterojunction thin films were fabricated by a spin-coating process on ITO-coated (thickness: 350 nm–400 nm) glass substrates (1.0 × 1.0 in$^2$, Hans Weidner GmbH, Nürnberg Germany). Pre-structured ITO coated glass substrates were subsequently cleaned in acetone and isopropyl alcohol for 10 min each. After drying, the substrates were bladed with 40 nm poly(3,4-ethylenedioxythiophene) polystyrene sulfonate (PEDOT:PSS, Clevios P VP Al 4083, Heraeus, Hanau, Germany). The DRCN5T films were spin-coated (1500 rpm) under inert gas atmosphere from the solutions of DRCN5T (purity ≥ 99%, 1-Material Inc., Dorval, Canada) and PC$_{71}$BM (purity ≥ 99%, Solenne B.V., Groningen, Netherlands) (1:0.8 wt.%) in chloroform leading to a film thickness of about 80 nm as estimated by a profilometer. The DRCN5T and PC$_{71}$BM solutions were stirred at 40 °C and 150 rpm before mixing and spin-coating. For the SVA post-processing procedure, the samples were loaded in the middle of a closed Petri dish containing 120 $\mu$l of respective solvent.

**Electron microscopy**. Energy filtered selected area electron diffraction experiments were performed on a ThermoFisher Scientific (TFS) monochromated, double $C_s$ corrected Titan Themis microscope operating at 300 kV, and NBD 4D-STEM and 4D-SCED on a TFS probe corrected Spectra microscope equiped with an X-CFEG gun and operated at 200 kV. The Themis is equiped with a regular Ceta (Scintillator coupled CMOS) camera capable to run at a maximum frame rate of 35 fps @ 512 × 512 pixels with dynamic range of about 13-bit (operated via TIA software), and the Spectra is coupled to a high sensitivity Ceta-S camera with maximum speed of 300 fps @ 512 × 512 pixels with dynamic range of about 12-bit (operated via Velox). For NBD, the optics were set to micro-probe STEM mode, with the smallest standard 50 $\mu$m C2 aperture and at the limit of convergence zoom. The smallest achievable convergence semi-angle are calibrated to be $\alpha = 0.7$ mrad (on the Themis) and $\alpha = 0.85$ mrad (on the Spectra), respectively, when the probe is focused at the eucentric height. On both microscopes, we noticed the factory alignment displays wrong convergence angle almost half of our calibrated values. To reach the confocal diffraction condition, basic column alignment of micro-probe STEM (at the smallest $\alpha$) were performed first, then diffraction lens was switched to imaging mode. Image magnification is first set to a high value, ~100 kx, to confirm that the confocal plane is properly reached, which is done by going through focus of the probe-defining (i.e, C3) lens and find the minimum size of the

probe. After this point, defocusing is achieved only done via moving sample up and down. The magnification of diffraction pattern will change upon shifting the sample, give full flexibility to tune/balance of the desired spatial and angular resolution. For a given defocus (i.e., sample offset) value, the image magnification is set so to appropriately cover the range of diffraction vectors of interest on the camera. The scan and de-scan pivot points were carefully aligned. Nevertheless, center beam shift of about 10–20 pixels (3–6 nm) were observed in SCED for a scanning field of view of about 2 $\mu$m (cf. Fig. S9). This is not critical and can be aligned precisely in post-acquisition processing. Finally the shape of probe is determined by the aberrations of the probe-forming lens, and distortions of the diffraction pattern can be corrected by carefully aligning the objective imaging lens (cf. Fig. S10).

**4D datasets acquisition due to dose limited resolution**. In all experiments, cameras were operated at their highest speed and other acquisition parameters (probe current, dwell time, defocus and scanning pixel distance) were estimated to keep both total and instaneous dose budget well below 10 $e^-$/Å$^2$. To estimate the dose limited resolution in a scanning probe experiment, we can use instaneous dose at any probing point

$$D\left[e^-/Å^2\right] = \frac{I_p \cdot 10^{-12}\left[\text{C/s}\right] \cdot 6.24 \cdot 10^{18}[e^-] \cdot t_{px} \cdot 10^{-3}[\text{s}]}{\pi \cdot \Delta R^2\left[Å^2\right]} \tag{3}$$

with $I_p$ probe current (in pA), $t_{px}$ dwell time at each pixel (in ms) and $D$ dose budget (in $e^-$/Å$^2$) that can be evaluated from SAED experiments, to be

$$\Delta R[Å] \approx 45\sqrt{I_p\left[\text{pA}\right] \cdot t_{px}[\text{ms}]/D\left[e^-/Å^2\right]}. \tag{4}$$

Considering the experiments performed on the Titan Themis platform, inserting the fastest camera frame time (roughly equals to the dwell time) $t_{px} = 13.5$ ms and dose bugest $D = 5\,e^-/Å^2$, a dose limited resolution of $\Delta R \approx 74$ Å is obtainable when setting probe current to $I_p = 1$ pA. Setting probe current too low will make searching area of interest difficult. With the smallest available convergence angle on our microscope under mirco-probe mode $\alpha = 0.7$ mrad, a defocus value $z > \Delta R/\alpha \approx 10\,\mu$m is needed. On the Spectra platform (smallest $\alpha = 0.85$ mrad) at the same dose budget of $D = 5\,e^-/Å^2$ and probe current of $I_p = 1$ pA, with shorter dwell time $t_{px} = 3.5$ ms, spatial resolution $\Delta R \approx 38$ Å is possible. To achieve this condition, a defocus value $z > \Delta R/\alpha \approx 4.5\,\mu$m is required. The scan pixel distance (sampling size) is set to slightly larger than the abovementioned resolution limit. For in situ experiments, it is important to balance spatial resolution required to reveal small structures in the early state, field of view to cover statistical relevant sample region, reasonable short time of data acquisition to minimize image distortion due to thermal drift. We took the following experimental parameters: 180 × 180 grid area with 4 nm STEM probe step size, defocus of 2.5 $\mu$m and probe current below the measurable quantity of 1 pA in the first few frames, and STEM probe step size increased to ~6 nm when large structured emerged and field of view was limited.

**Data handling, pre-processing and visualization**. The acquired data were pre-processed in Gatan DigitalMicrograph software with public plugins and homemade scripts. The acquired data is firstly aligned to account for the shift of center beam. Elliptical distortion of the sum diffraction pattern was observed in some of the SCED experiments, which can be hardware corrected using the objective lens stigmator (cf. Fig. S10). The distortion was observed to not depend on the scanning position when the scanning field is within few micrometers. This makes the post-acquisition software correction of individual pattern feasible based on the distortion evaluated from the summed patterns. For this, we applied the evaluation and correction method developed by Mitchell and Van den Berg[45]. Due to the high SNR in the SCED datasets, the diffraction signals at small angles were obvious and straightforward for subsequent analysis using simple virtual apertures; and no hardware elastic filtering (like in Fig. 3b) or post-processing background subtraction[46] were necessary. The 2D orientation map of edge-on crystal domains was obtained by directly evaluating the virtual dark-field image intensities using sector apertures of radius between 2.6 and 2.8 nm$^{-1}$ and angular increment of $n$ degrees. In this way a, total of $360/n$ dark-field images were calculated, and each image corresponds to in-plane diffraction angle $\phi = ni$ defined by the sequence $i$ of the virtual apertures. This reduces the 4D dataset to a 3D dark-field image data cube. At any (real space) scanning pixels, we search for the maximum intensity $M$ along the dark-field sequence and mark its location $\phi$, which together defines 2D grid of complex numbers $M(x,y)e^{i\phi(x,y)}$ (equivalent to 2D vectors). For face-on crystallites or overlapping crystallites, multiple diffraction spot may appear at same diffraction angle $q$, and the above algorithm fails to capture same orientation crystal domains with slight out-of-plane tilt (which result in change of the center of Lauer circle and flicking of diffraction spots intensities). For this, we apply a rotation invariant template matching method to find location of face-on domains and determine its rotation angle. Here, the rotation invariant template matching is converted to a shift-invariant template matching via cross-correlation of the polar transformed

diffraction patterns. Finally, the processed data in form of 2D vectors were visualized using Python with Matplotlib, OpenCV and numpy.

## Data availability

All electron microscopy data generated in this study have been deposited in open-access data repository ZENODO under accession code https://doi.org/10.5281/zenodo.5831202. Source data are provided with this paper.

## Code availability

The DigitalMicrograph scripts for data analysis and Python codes for data visualization are supplied in SI as a zip file.

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

## Acknowledgements

We thank Dr. Lilian Vogl for providing the Au/Pt sample (Fig. S9), and Dr. Dave R. Mitchell for help implementing the elliptical correction work flow using DM-scripting. We acknowledge financial support from Deutsche Forschungsgemeinschaft (DFG) via the research training school GRK 1896: "In-Situ Microscopy with Electrons, X-rays and Scanning Probes", the Cluster of Excellence EXC 315 "Engineering of Advanced Materials" and SFB 953 "Syntetic Carbon Allotropes". R.H.F. acknowledges financial support by the Bundesministerium für Bildung und Forschung (contract 05K19WE2). Work at the Molecular Foundry was supported by the Office of Science, Office of Basic Energy Sciences, of the U.S. Department of Energy under Contract No. DE-AC02-05CH11231.

## Author contributions

M.W. and E.S. conceived the idea and designed the experiments. C.H. prepared the samples and performed extensive TEM characterization of the DRCN5T:PCBM samples. M.J. prepared the $\alpha, \omega$-DH6T samples under supervision of R.H.F. M.W. and C.H. performed the in situ experiments. M.W. implemented 4D-SCED experimentally, developed algorithms to analyze the data, and drafted the manuscript with input from all authors. C.O. examined the 4D datasets to verify the results using alternative analysis and visualization routines. All authors have given approval to the final version of the manuscript.

## Funding

## Competing interests

The authors declare no competing interests.
