## [Peer Review File · Nature Communications]

Reviewers' Comments:

Reviewer #1:

Remarks to the Author:

This is an excellent paper that presents an innovative approach to obtaining local structural information with low dose and good angular resolution using a transmission electron microscope in scanning mode. The paper demonstrates the power of the method by applying it to some challenging organic solar cell materials enabling important features in these materials to be observed for the first time (at least as far as I am aware). The paper is well-constructed and well-written, (apart for being riddled with English errors).

The paper presents a creative, new and useful TEM method and, as such, is very suitable for Nature Communications. However, before publication I would recommend the authors address the following minor points (in no particular order):

- It is promising that the method can be applied to in-situ heating experiments, albeit at relatively low spatial resolution. The ultimate usefulness of this approach for in-situ work depends on multiple factors. It would be helpful if the authors could add a paragraph expanding on the potential, and also the technical limits, of using this method for in-situ work.
- Can the authors comment in the manuscript on what constraints, if any, the precision of the descans coils place on the data quality in this configuration?
- Can the authors comment in the manuscript on the extent to which chromatic aberration in the post-specimen lenses might be a problem for this technique?
- Line 47: The authors say: "The distribution of intensity in the beam disks does not typically contain additional information,..."

This statement is not strictly true as written. I think the authors may mean something a little different and should rephrase this statement. The angular distribution DOES contain significant information it is just this may not be useful in the context of the aim of this experiment.

- Line 51: The authors say: "reduces the angular resolution because of disk overlap"

This statement is also not strictly true as written. Again, I think this just needs rephrasing for clarity. Disk overlap in itself, does not reduce angular resolution within the disk. Again, I think the authors mean something else but it reads as if they are referring to angular resolution within the disk and this may confuse some readers.

- Line 261 and surrounding discussion:

I largely agree with the author's assessment, however it might be prudent for the authors to mention caveats such as excitation errors, which could be slightly different between the measurements.

- Line 325 and surrounding discussion:

This sentence about wavefunction overlapping is not clear, as the term "overlapping" is used to describe interference within the TEM wave throughout the paper (such as between Bragg discs). Here I think the authors are referring to the directional anisotropy of possible charge transfer within the material as being likely in-plane due to the pi-conjugation and stacking configuration. They should consider rewording to make this point clearer (also paper [16] concerns strain mapping techniques, and so does not help to clarify this sentence).

- Line 329 and surrounding discussion:

This interpretation of the charge carrier transfer is reminiscent of that in graphite fibres. If this is the case, then the authors could add clarity here to note the similarity and add an appropriate

reference.

- Figure 4, 5:

What is causing the streaking in the 'orange' data (presumably charging)? This should be discussed in the main text (I didn't see any comment on this).

- One of the key outcomes of the method presented here is to avoid disk overlap. Given the importance of this to this paper, the authors should place this in context. While the authors approach is distinct and has different aims, they should refer to existing methods that achieve this:

e.g.

Michiyoshi TANAKA, Ryuichi SAITO, Katsuyoshi UENO, Yoshiyasu HARADA, Large-Angle Convergent-Beam Electron Diffraction, *Journal of Electron Microscopy*, Volume 29, Issue 4, 1980, Pages 408–412, <https://doi.org/10.1093/oxfordjournals.jmicro.a050262>

Jean-Paul Morniroli

Textbook entitled: Large-Angle Convergent-Beam Electron Diffraction Applications to Crystal Defects

Christoph T Koch

Ultramicroscopy 2011 111(7):828-40. doi: 10.1016/j.ultramic.2010.12.014
Aberration-compensated large-angle rocking-beam electron diffraction

R. Beanland, P. J. Thomas, D. I. Woodward, P. A. Thomas and R. A. Roemer

Digital electron diffraction - seeing the whole picture

Acta Cryst. (2013). A69, 427-434

<https://doi.org/10.1107/S0108767313010143>

- Similarly, a key aspect of this paper is the operation in a confocal-type mode. Again, the authors approach is distinct and has different aims, but it is helpful to the reader if they can elaborate on its relationship to existing confocal TEM methods and expand on the references:

e.g.

P. D. Nellist, G. Behan, A. I. Kirkland and C. J. D. Hetherington

"Confocal operation of a transmission electron microscope with two aberration correctors",

Appl. Phys. Lett. 89, 124105 (2006) <https://doi.org/10.1063/1.2356699>

Peng Wang, Gavin Behan, Masaki Takeguchi, Ayako Hashimoto, Kazutaka Mitsuishi, Masayuki Shimojo, Angus I. Kirkland, and Peter D. Nellist

Nanoscale Energy-Filtered Scanning Confocal Electron Microscopy Using a Double-Aberration-Corrected Transmission Electron Microscope

Phys. Rev. Lett. 104, 200801; DOI: 10.1103/PhysRevLett.104.200801

D'Alfonso AJ, Cosgriff EC, Findlay SD, Behan G, Kirkland AI, Nellist PD, Allen LJ.

Three-dimensional imaging in double aberration-corrected scanning confocal electron microscopy, part II: inelastic scattering. *Ultramicroscopy*. 2008 Nov;108(12):1567-78. doi:

10.1016/j.ultramic.2008.05.007.

Joanne Etheridge, Sorin Lazar, Christian Dwyer, and Gianluigi A. Botton

Imaging High-Energy Electrons Propagating in a Crystal

Phys. Rev. Lett. 106, 160802; DOI: 10.1103/PhysRevLett.106.160802

Changlin Zheng, Ye Zhu, Sorin Lazar, and Joanne Etheridge

Fast Imaging with Inelastically Scattered Electrons by Off-Axis Chromatic Confocal Electron Microscopy; *Phys. Rev. Lett.* 112, 166101; doi.org/10.1103/PhysRevLett.112.166101

And possibly the following, though less relevant:

Andreas Rosenauer, Florian F. Krause, Knut Müller, Marco Schowalter, and Thorsten Mehrrens
Conventional Transmission Electron Microscopy Imaging beyond the Diffraction and Information
Limits: Phys. Rev. Lett. 113, 096101: <https://doi.org/10.1103/PhysRevLett.113.096101>

S. Lazar, J. Etheridge, C. Dwyer, B. Freitag and G. A. Botton
Atomic resolution imaging using the real-space distribution of electrons scattered by a crystalline
material Acta Cryst. (2011). A67, 487-490, <https://doi.org/10.1107/S0108767311020708>

- Line 88:

At this point (line 88), it is worth citing one or more papers on the CTEM 'spot scan' method, which was motivated by low-dose considerations. This is essentially small-convergence angle CTEM with a small spot that is scanned to produce low-dose and highly coherent phase contrast images.

e.g. Kenneth H. Downing,
Spot-Scan Imaging in Transmission Electron Microscopy
Science 251 (1991) 53-59. DOI: 10.1126/science.1846047

- Line 119: The authors suggest this method can be executed on a "standard instrument".

The method needs descan coils. This is not standard in many instruments, though it is on the systems the authors happen to be using. Suggest modifying the text accordingly.

- Line 130: "stable confocal plane" – what does "stable" mean here?

- Line 167: What does "suppress the diffraction information between the primary beam and the diffraction beam" mean?

- There were multiple typos and errors in English usage which made me concerned as to whether the native English speakers in the authorship had read the manuscript! Here is just a sample that I picked up but the manuscript needs a careful reading to correct these issues and other English usage issues.

Line 52: 'study' should be 'studying'

Line 53: typically a few nanometres

Figure 1 caption: there are some missing definite articles and in line (c) some superfluous hypens

Line 77: 'lens' should be 'lenses'

Line 106: two 'well's in the one sentence. Suggest lens aberration can be "ignored"

Line 156: should be 'the confocal setup'

Line 179: "a smaller convergence"

Line 197: "conditions"

Line 215, 293: inconsistent acronym: GIWAX or GIWAXS?

Figure 3, 4: more misplaced hypens

Line 265: should be "will be"

Line 253: Figures – should be plural.

Line 337: "Figures" "show" or "Figure" "shows"

Line 344: plural and tense mix-up: "samples have" or "sample has"

Line 360: should be "seems"

Line 365: should be "hints"

Reviewer #2:

Remarks to the Author:

The manuscript describes the use of a confocal arrangement to enable low-dose electron diffraction work from a very beam sensitive organic sample. The authors duly note the similarity of the approach to the large angle convergent beam electron diffraction method and bright-field

scanning confocal electron microscopy and the degree of innovation in the microscope configuration used is limited. Using this approach to allow for diffraction from this challenging sample is, however, an impressive piece of work. The method is used to examine the orientation of nanocrystallites and their structural evolution on thermal annealing. I think it will be of wide interest and is suitable for publication in Nature Communications.

In the trade-off between spatial resolution and the resolution (and therefore signal to noise) in the diffraction pattern, the limit is controlled by diffraction (or equivalently the Heisenberg uncertainty principle), and no method can beat this fundamental limit. In the work, the authors are achieving a specific angular resolution in their diffraction pattern, and are illuminating a particular region of the sample. Such conditions could also be achieved with a very low convergence angle in conventional diffraction. The authors make the point that achieving such low convergence angles is too challenging from a practical point of view, hence the need for the confocal approach. It would be useful to explain that point a little more to more strongly justify the confocal approach. What would be the exact convergence angle needed to match the confocal work, and why would that make the experiment intractable?

Overall the manuscript is well written. Other more minor comments:

The citing of the references is not in numerical order.

The suggestion on line 48 of page 1 that spreading the discs lowers the S/N is only true if the DQE of the detector is significantly below 1. For a direct electron detector, no S/N loss should occur, and spreading the diffracted beams may help limit saturation.

Reviewer #3:

Remarks to the Author:

Wu et al. introduced a new 4D scanning confocal electron diffraction technique and use it to study the structure of molecular nano-crystallites in the DRCN5T:PC71BM bulk heterojunction thin films processed by thermal annealing and solvent vapor annealing treatments. As compared to the standard NBD setup, this new technique can effectively reduce dose by about an order of magnitude, and also enables in situ monitoring of structural evolution of nano-crystallites at elevated temperatures. This work is very interesting, can provide a new opportunity to study the bulk microstructure of donor:acceptor mixed blend films. I suggest that this manuscript can be accepted with major revisions. Some important comments are provided below.

1. The molecular structure of the investigated materials should be provide in Figure 2. The present representation of the DFT structure does not clearly guide the chemical structures of DRCN5T and PC71BM.
2. In Figure 4d, the location of the acceptor is not so clear. Can the authors provide the rational explanations?
3. in order to demonstrate this feasibility of this technique on investigating the bulk microstructure of the BHJ blend, the relevant morphological characteristics, such as AFM and GIWAXS measurements, should be provided.
4. About the thermal annealing treatments, the images tested by 4D-SCED show the structural evolution of the active layer during annealing at elevated temperatures, with small rod-like shape in projection. However, this result is different with the previous results reported in the literature. The authors should explain the different phenomenon.
5. To illustrate the merit of this technique, some other soft materials should be introduced to analyze the nano-crystallography.
6. Why the length of the scale bars in Figure 5f is shorter than the other images? For clear comparisons, the scale bars should be the same.
7. The writing of this manuscript is too colloquial. It is suggested to explain related ideas and results through the third person perspective.

Point-by-point response to comments of reviewer

Reviewer #1 (Remarks to the Author):

This is an excellent paper that presents an innovative approach to obtaining local structural information with low dose and good angular resolution using a transmission electron microscope in scanning mode. The paper demonstrates the power of the method by applying it to some challenging organic solar cell materials enabling important features in these materials to be observed for the first time (at least as far as I am aware). The paper is well-constructed and well-written, (apart for being riddled with English errors). The paper presents a creative, new and useful TEM method and, as such, is very suitable for Nature Communications.

However, before publication I would recommend the authors address the following minor points (in no particular order):

We sincerely thank the reviewer for his/her very detailed evaluation of our work and helpful and constructive comments to improve our manuscript.

- It is promising that the method can be applied to in-situ heating experiments, albeit at relatively low spatial resolution. The ultimate usefulness of this approach for in-situ work depends on multiple factors. It would be helpful if the authors could add a paragraph expanding on the potential, and also the technical limits, of using this method for in-situ work.

We appreciate the comment. There are indeed many factors affecting in situ observations of structural evolution.

The most important aspect is the dose budget of sample at the relevant conditions. The electron beam induces permanent, irreversible impact (e.g., generation of charged states, radicals, etc.), even at vanishingly low dose, which may influence the kinetics. In our experiment, regions with illumination history were “dead” and no further growth or ripening would follow at later stage of temperature and time. Below critical dose, the beam freezes the kinetics thus no growth of structure (from top to down, which took ~2 min.) was observed in any of the single frame, but still allow the structure at the first light to be probed. Move to fresh sample regions is necessary to record the true structure evolution. Finally, quantitative knowledge gain can only be drawn from in situ observations if data are validated via ex situ (interrupted) studies. The 4D-SCED setup optimizes the ratio of structure information to dose for such samples and the parameters should be chosen to accommodate the available dose budget and required spatial resolution. In the current work, scintillator coupled CMOS cameras were applied. Use of state-of-the-art direct detection cameras with DQE approaching 1 and higher frame rates is expected to further improve the spatial resolution (cf. methods section).

We have included an additional paragraph in the revised manuscript to account for this.

- Can the authors comment in the manuscript on what constraints, if any, the precision of the descanner coils place on the data quality in this configuration?

Since 4D datasets are recorded, the requirement on the precision of the descanned image is not so strict. Residual image shifts can be easily corrected by numerical post-acquisition alignment, as long as the diffraction space field of interest still fits in the camera field of view. Allowing the center beam to shift over a few pixels also helps not to burn out the camera. We added such a statement in the materials and methods section (highlighted)

- Can the authors comment in the manuscript on the extent to which chromatic aberration in the post-specimen lenses might be a problem for this technique?

Chromatic aberration will cause a defocus spread of the diffraction spots and which slightly reduces the diffraction signal to background, as considered in detail in the following:

In the measured EELS spectra of the samples, the log-ratio thickness is typically 0.4-0.6, and the zero-loss peak is about 2 orders of magnitude higher than the plasmon peak and more than 4 orders of magnitude higher than the carbon K-edge. Taking a typical chromatic aberration of $C_c=2.5$ mm and primary beam energy of 200keV used in this work, the chromatic defocus of electrons undergone plasmon losses (peaked at ~ 25 eV) is $df = C_c \cdot \Delta E / E_0 \sim 350$ nm, and ~ 3.5 μ m for carbon (the major constituent in BHJs), which is in the same order of defocus applied to the sample in the current study. So long as the illumination is on axis, which is valid in our SCED setup, the chromatic defocus will cause blurred disks superimposed concentrically to the diffraction spots, and thus do not influence the accuracy to locate the Bragg spots and structural studies. Nonetheless, this will reduce the diffraction signal to background ratio, which depends on the sample thickness.

- Line 47: The authors say: “The distribution of intensity in the beam disks does not typically contain additional information,...”

This statement is not strictly true as written. I think the authors may mean something a little different and should rephrase this statement. The angular distribution DOES contain significant information it is just this may not be useful in the context of the aim of this experiment.

Thanks for pointing out. The intensity inside the beam disks is the basis for retrieval of structural information in e.g., CBED and ptychography etc., but not for “simple” crystallography-related studies relying only on the geometric position of diffraction spots. We have rephrased that in the revised version.

- Line 51: The authors say: “reduces the angular resolution because of disk overlap”

This statement is also not strictly true as written. Again, I think this just needs rephrasing for clarity. Disk overlap in itself, does not reduce angular resolution within the disk. Again, I think the authors mean something else but it reads as if they are referring to angular resolution within the disk and this may confuse some readers.

We agree with the reviewer about the confusion in using angular resolution. We thus clearly defined/restrained the usage of “angular resolution” in the revised version now referring to the accuracy of locating Bragg diffraction disks/spots.

- Line 261 and surrounding discussion:

I largely agree with the author's assessment, however it might be prudent for the authors to mention caveats such as excitation errors, which could be slightly different between the measurements.

The illumination conditions in the comparison experiments are identical (convergence angle α and sample defocus z) as well as the dwell time thus leading to identical dose conditions. We changed only the lenses at the imaging side. Therefore, the excitation errors should be the same, only that this information is recorded in the intensity distribution (i.e., rocking curve) in the disks in NBD setup, but not in the SCED setup.

- Line 325 and surrounding discussion:

This sentence about wavefunction overlapping is not clear, as the term "overlapping" is used to describe interference within the TEM wave throughout the paper (such as between Bragg discs). Here I think the authors are referring to the directional anisotropy of possible charge transfer within the material as being likely in-plane due to the pi-conjugation and stacking configuration. They should consider rewording to make this point clearer (also paper [16] concerns strain mapping techniques, and so does not help to clarify this sentence).

We thank the reviewer for pointing this out. Indeed, the statement and discussion there refers to highly anisotropic charge carrier transport in such flat pi-conjugated systems. The original citation [16] was wrongly linked. It is corrected now to refer to the correct paper that discuss the charge carrier transport properties.

Line 329 and surrounding discussion:

This interpretation of the charge carrier transfer is reminiscent of that in graphite fibres. If this is the case, then the authors could add clarity here to note the similarity and add an appropriate reference.

Yes, it is improved for clarity in the revised version.

- Figure 4, 5:

What is causing the streaking in the 'orange' data (presumably charging)? This should be discussed in the main text (I didn't see any comment on this).

The streaking of the acceptor maps does not match that of the streaking in ADF as well the calculated bright field images. The later two agree, which is known to us as consequence of the intensity fluctuation from the cold FEG (flickering noise). The disagreement as well more prominent streaking in the in situ experiment indeed points to a charging effect.

We added a short comment about this in the revised manuscript.

- One of the key outcomes of the method presented here is to avoid disk overlap. Given the importance of this to this paper, the authors should place this in context. While the authors approach is distinct and has different aims, they should refer to existing methods that achieve this:

e.g.

Michiyoshi TANAKA, Ryuichi SAITO, Katsuyoshi UENO, Yoshiyasu HARADA, Large-Angle Convergent-Beam

Electron Diffraction, Journal of Electron Microscopy, Volume 29, Issue 4, 1980, Pages 408–412,
<https://doi.org/10.1093/oxfordjournals.jmicro.a050262>

Jean-Paul Morniroli

Textbook entitled: Large-Angle Convergent-Beam Electron Diffraction Applications to Crystal Defects

Christoph T Koch

Ultramicroscopy 2011 111(7):828-40. doi: 10.1016/j.ultramic.2010.12.014

Aberration-compensated large-angle rocking-beam electron diffraction

R. Beanland, P. J. Thomas, D. I. Woodward, P. A. Thomas and R. A. Roemer

Digital electron diffraction - seeing the whole picture

Acta Cryst. (2013). A69, 427-434

<https://doi.org/10.1107/S0108767313010143>

Thanks for the suggestion, these literature references are included in the revised version.

• Similarly, a key aspect of this paper is the operation in a confocal-type mode. Again, the authors approach is distinct and has different aims, but it is helpful to the reader if they can elaborate on its relationship to existing confocal TEM methods and expand on the references:

e.g.

P. D. Nellist, G. Behan, A. I. Kirkland and C. J. D. Hetherington

"Confocal operation of a transmission electron microscope with two aberration correctors",

Appl. Phys. Lett. 89, 124105 (2006) <https://doi.org/10.1063/1.2356699>

Peng Wang, Gavin Behan, Masaki Takeguchi, Ayako Hashimoto, Kazutaka Mitsuishi, Masayuki Shimojo, Angus I. Kirkland, and Peter D. Nellist

Nanoscale Energy-Filtered Scanning Confocal Electron Microscopy Using a Double-Aberration-Corrected Transmission Electron Microscope

Phys. Rev. Lett. 104, 200801; DOI: 10.1103/PhysRevLett.104.200801

D'Alfonso AJ, Cosgriff EC, Findlay SD, Behan G, Kirkland AI, Nellist PD, Allen LJ.

Three-dimensional imaging in double aberration-corrected scanning confocal electron microscopy, part II: inelastic scattering. Ultramicroscopy. 2008 Nov;108(12):1567-78. doi: 10.1016/j.ultramic.2008.05.007.

Joanne Etheridge, Sorin Lazar, Christian Dwyer, and Gianluigi A. Botton

Imaging High-Energy Electrons Propagating in a Crystal

Phys. Rev. Lett. 106, 160802; DOI: 10.1103/PhysRevLett.106.160802

Changlin Zheng, Ye Zhu, Sorin Lazar, and Joanne Etheridge

Fast Imaging with Inelastically Scattered Electrons by Off-Axis Chromatic Confocal Electron Microscopy;

Phys. Rev. Lett. 112, 166101; doi.org/10.1103/PhysRevLett.112.166101

And possibly the following, though less relevant:

Andreas Rosenauer, Florian F. Krause, Knut Müller, Marco Schowalter, and Thorsten Mehrtens
Conventional Transmission Electron Microscopy Imaging beyond the Diffraction and Information Limits:
Phys. Rev. Lett. 113, 096101: <https://doi.org/10.1103/PhysRevLett.113.096101>

S. Lazar, J. Etheridge, C. Dwyer, B. Freitag and G. A. Botton
Atomic resolution imaging using the real-space distribution of electrons scattered by a crystalline
material Acta Cryst. (2011). A67, 487-490, <https://doi.org/10.1107/S0108767311020708>

We thank the reviewer for the extended literature in confocal electron microscopy. It does indeed lay down a solid background of SCEM for readers. We have included them at appropriate context and some with comments in the revised version (highlighted).

• Line 88:

At this point (line 88), it is worth citing one or more papers on the CTEM ‘spot scan’ method, which was motivated by low-dose considerations. This is essentially small-convergence angle CTEM with a small spot that is scanned to produce low-dose and highly coherent phase contrast images.

e.g. Kenneth H. Downing,

Spot-Scan Imaging in Transmission Electron Microscopy
Science 251 (1991) 53-59. DOI: 10.1126/science.1846047

It is now included.

• Line 119: The authors suggest this method can be executed on a “standard instrument”.

The method needs descans coils. This is not standard in many instruments, though it is on the systems the authors happen to be using. Suggest modifying the text accordingly.

The statement is removed in the revised version.

• Line 130: “stable confocal plane” – what does “stable” mean here?

We imply that the physical defocus of the sample will not cause the focal point (spot diffraction patterns) to shift along the z-axis, thus cause disk pattern. This might be obvious or ambiguous in the statement, we therefore removed this word in the revised version.

• Line 167: What does “suppress the diffraction information between the primary beam and the diffraction beam” mean?

Since the diffraction spots are not identical to that of true far field pattern (e.g., that obtained using parallel beam illumination in SAED), the spread of the diffraction spots in SCED depends on its propagation distance z. We want to suppress this spread of the diffraction spots, and its potential overlap/interference with other spots.

We changed it to “suppress the spread of the diffraction spots” in the revised version.

- There were multiple typos and errors in English usage which made me concerned as to whether the native English speakers in the authorship had read the manuscript! Here is just a sample that I picked up but the manuscript needs a careful reading to correct these issues and other English usage issues.

Line 52: ‘study’ should be ‘studying’

Line 53: typically a few nanometres

Figure 1 caption: there are some missing definite articles and in line (c) some superfluous hypens

Line 77: ‘lens’ should be ‘lenses’

Line 106: two ‘well’s in the one sentence. Suggest lens aberration can be “ignored”

Line 156: should be ‘the confocal setup’

Line 179: “a smaller convergence”

Line 197: “conditions”

Line 215, 293: inconsistent acronym: GIWAX or GIWAXS?

Figure 3, 4: more misplaced hyphens

Line 265: should be “will be”

Line 253: Figures – should be plural.

Line 337: "Figures" "show" or "Figure" "shows"

Line 344: plural and tense mix-up: "samples have" or "sample has"

Line 360: should be “seems”

Line 365: should be “hints”

We thank the reviewer for such a detailed scrutiny of the manuscript. As submitting author, I apologize for the typos which entered at the last step of the initial submission when I decided to convert the word file, which all authors approved, to LateX. These errors are all corrected in the revised manuscript (with highlights of changes).

Reviewer #2 (Remarks to the Author):

The manuscript describes the use of a confocal arrangement to enable low-dose electron diffraction work from a very beam sensitive organic sample. The authors duly note the similarity of the approach to the large angle convergent beam electron diffraction method and bright-field scanning confocal electron microscopy and the degree of innovation in the microscope configuration used is limited. Using this approach to allow for diffraction from this challenging sample is, however, an impressive piece of work. The method is used to examine the orientation of nanocrystallites and their structural evolution on thermal annealing. I think it will be of wide interest and is suitable for publication in Nature Communications.

In the trade-off between spatial resolution and the resolution (and therefore signal to noise) in the diffraction pattern, the limit is controlled by diffraction (or equivalently the Heisenberg uncertainty principle), and no method can beat this fundamental limit. In the work, the authors are achieving a specific angular resolution in their diffraction pattern and are illuminating a particular region of the sample. Such conditions could also be achieved with a very low convergence angle in conventional diffraction. The authors make the point that achieving such low convergence angles is too challenging from a practical point of view, hence the need for the confocal approach. It would be useful to explain that point a little more to more strongly justify the confocal approach. What would be the exact convergence angle needed to match the confocal work, and why would that make the experiment intractable?

We appreciate the detailed evaluation of our work.

The whole point for the study of complex nano-crystallography in soft material is to obtain spot-like diffraction signals (good SNR) while keeping a spatial resolution (small illumination area) relevant for the feature of interest. The benefit of confocal setup is that (1) we form a pencil beam on the sample (which has roughly circular profile) but can still focus down the diffraction to sharp spots, and (2) the probe spatial extent in diffraction space is largely decoupled from the convergence angle (only through Airy disc).

We have rephrased the corresponding section and highlighted these changes in the revised version.

Overall the manuscript is well written. Other more minor comments:

The citing of the references is not in numerical order.

This is corrected in the revised version.

The suggestion on line 48 of page 1 that spreading the discs lowers the S/N is only true if the DQE of the detector is significantly below 1. For a direct electron detector, no S/N loss should occur, and spreading the diffracted beams may help limit saturation.

We agree that the loss of SNR is only true if the DQE of the detector is significantly below 1 merely from consideration of detector noise and spread the beams help limit saturation of detector. In reality, the elastically scattered diffraction signals of interest at small angles are heavily affected by the strong inelastically scattered background (which we appropriately generalized as noise). Focusing the disk to spot help to lift the elastic diffraction above the inelastic background. Although we could not perform

this comparing 4D-NBD and 4D-SCED with elastic filtering, this is very obviously seen when we compare the diffraction signals in SAED patterns with and without elastic filtering shown below.

Selection area electron diffraction (SAED) patterns of a ~80 nm thin film of DRCN5T:PC71BM blends after SVA in CS2 for 840s, with (above) and without (below) filtering the elastically scattered electrons using a 10eV energy selection slit around the zero-loss peak. Azimuth integrated intensity profiles are show to the right side. These diffraction patterns are taken from fresh area of the same sample at total dose below $1e^{-}/\text{\AA}^2$.

In this sense, the term SNR is not appropriate, we should rather refer to signal to background.

At higher scattering angles, e.g., the pi-stacking diffraction spots, the intensity is very weak and the signals in the corresponding NBD disks can become sparse at very low dose. Even if we would be able to detect the signals using a DD camera in NBD setup this is expected to reduce the accuracy of determining the exact location of diffraction spot thus lowering the angular resolution (dose-limited angular resolution). Focusing the disks to spots helps to concentrate the signals, resulting in better angular resolution.

We acknowledge this important comment and improved the statement regarding this point in the revised version of manuscript.

Reviewer #3 (Remarks to the Author):

Wu et al. introduced a new 4D scanning confocal electron diffraction technique and use it to study the structure of molecular nano-crystallites in the DRCN5T:PC71BM bulk heterojunction thin films processed by thermal annealing and solvent vapor annealing treatments. As compared to the standard NBD setup, this new technique can effectively reduce dose by about an order of magnitude, and also enables in situ monitoring of structural evolution of nano-crystallites at elevated temperatures. This work is very interesting, can provide a new opportunity to study the bulk microstructure of donor:acceptor mixed blend films. I suggest that this manuscript can be accepted with major revisions. Some important comments are provided below.

We are grateful to the comments provided by the reviewer from materials science perspective. We have addressed most of the comments and request in the revised manuscript as listed in the following.

1. The molecular structure of the investigated materials should be provide in Figure 2. The present representation of the DFT structure does not clearly guide the chemical structures of DRCN5T and PC71BM.

We thank the reviewer for pointing this. The molecular structure is replaced with chemical representation as suggested.

2. In Figure 4d, the location of the acceptor is not so clear. Can the authors provide the rational explanations?

In this figure, the intensity of diffraction signal of PC71BM is represented using the “black body” color representation. The brighter (brown to yellow to white) the intensity means more PC71BM diffraction signals at the given pixel location. In Fig. 4d, the enrichment of PC71BM is clearly seen close to the interface to the edge-on DRCN5T crystallite rods (by comparing with Fig. 4c). The comment of the reviewer might be related to the unclearly defined color intensity representation of data in different community. A color intensity bar is added in the revised version.

3. in order to demonstrate this feasibility of this technique on investigating the bulk microstructure of the BHJ blend, the relevant morphological characteristics, such as AFM and GIWAXS measurements, should be provided.

We totally understand the concern raised by the reviewer. We fully agree on the importance of correlative characterization of BHJ to render a complete structural and morphology understanding. Indeed, we have extensively and systematically studied the nano-morphology and structure character using correlative GIWAXS, electron diffraction and energy-filtered TEM or STEM-EELS elemental mapping. (1) The GIWAXS data of two of the similarly processed samples used in this work (the samples after SVA in CS₂ and in CHCl₃) have been published in an earlier paper by Berlinghof et. al. (ref [22]), and we believe also properly mentioned in the manuscript. The structural picture derived using 4D-SCED in this work is in accordance with that from GIWAXS but with much more insight in spatial and temporal dimensions. (2) To elucidate the nano-morphology, we have included the elemental maps of carbon and

sulfur of the identical samples studied in this work using STEM-EELS, an established analytical TEM method. Since the DRCN5T donor is sulfur-rich and acceptor PC71BM is carbon-rich, these elemental maps provide unambiguous view of the nano-morphology which is more reliable than topology measurement using AFM as suggested by the reviewer. We would like to emphasize, although STEM-EELS or EFTEM typically require a dose several orders of magnitude higher than the critical dose of total structural damage to deliver enough contrast, nano-morphology of BHJ thin films can still be routinely revealed at nanoscale if structural damage, i.e. damage of the crystalline order, does not cause delocalization of the constituent elements, which is the case in most BHJ thin films.

4. About the thermal annealing treatments, the images tested by 4D-SCED show the structural evolution of the active layer during annealing at elevated temperatures, with small rod-like shape in projection. However, this result is different with the previous results reported in the literature. The authors should explain the different phenomenon.

We totally understand the concern raised by the reviewer. We believe that the difference to literature reports is mainly due to the incomplete (structural) information gained via the different characterization tools used in the respective literature. AFM, TEM (bright field, high resolution imaging as well energy-filtered TEM) are the routine methods applied in literature to reveal the nanomorphology of BHJs. However, no local structural (i.e., nanoscale crystalline and orientation) information is obtained with these characterization methods, both due to the contrast mechanism itself and because the dose required to obtain meaningful contrast is orders of magnitude higher than the critical dose of structural damage. Since the reviewer did not mention the exact literature, we discuss in the following based on the following literature:

[1] Min, J., Jiao, X., Sgobba, V., Kan, B., Heumüller, T., Rechberger, S., ... Brabec, C. J. (2016). High efficiency and stability small molecule solar cells developed by bulk microstructure fine-tuning. *Nano Energy*, 28, 241–249. <https://doi.org/10.1016/j.nanoen.2016.08.047>

[2] Zhang, Q., Kan, B., Liu, F., Long, G., Wan, X., Chen, X., ... Chen, Y. (2014). Small-molecule solar cells with efficiency over 9%. *Nature Photonics*, 9(1), 35–41. <https://doi.org/10.1038/nphoton.2014.269>

While the identical material system has been studied in [1], a very similar material system (DRCN7T:PC71BM) has been reported in [2]. Nanomorphology of the blends after thermal annealing under 120C for 10 min (similar to, but not identical to condition in Fig. 5d) has been reported in [1]. And TEM image shown in [2] is a DRCN7T:PC71BM sample thermally annealed for 10min under an unclearly stated temperature.

In ref [1], the nano-morphology has been studied using with AFM and EFTEM imaging. A “bi-continuous interpenetrating network” structure has been interpreted and reported based on the EFTEM image contrast. Due to the projection effect in EFTEM and the lack of crystal orientation information, the differently orientated domains in projection results in a seemingly interconnected continuous donor structure. In ref [2] bright field TEM imaging is applied to reveal the morphology of DRCN7T:PC71BM, after annealing, “a network of fibrils with diameter of ~10 nm” is interpreted based on the image contrast. The contrast in BF TEM of OSC are mostly from mass-thickness, diffraction contrast if dose and damage would be carefully controlled, and different scattering potentials at interfaces (in the defocused, i.e., in-line holography, images). Lacking structural contrast and wrapped with projection problem, the interpreted structural character can be incomplete.

With the additional information we gained using 4D-SCED, the individual donor crystalline domains are clearly revealed. They are already separated domains from the very beginning of freshly coated film. Actually, a similar “interconnected and continuous” picture might be wrongly interpreted if we inspect only the simultaneously acquired STEM-ADF images, which is dominated by mass-thickness contrast. The contrast of these images even match EFTEM images of our ex situ TA experiments. To concretely elucidate this, we have included the ADF images in Fig. 5 and few sentence discussion around it. In addition, our nanomorphology studies of thin film blends using EFTEM (carbon signals) after ex situ annealing under various temperatures in SI and added a short comment on this, these changes are highlighted in the main manuscript.

5. To illustrate the merit of this technique, some other soft materials should be introduced to analyze the nano-crystallography.

The material system (DRCN5T:PCBM) we chose in this work represent a very challenging one in terms of dose budget and required spatial resolution. As presented in our study, the critical dose of total damage of pi-stacking is only around $5e^{-}/\text{\AA}^2$ under RT. The reported critical dose of for example, P3TH:PCBM, a well-studied organic solar cell material system, is about $16 - 19 e^{-}/\text{\AA}^2$ (Leijten, Z. J. W. A., Keizer, A. D. A., De With, G., & Friedrich, H. (2017). Quantitative Analysis of Electron Beam Damage in Organic Thin Films. *Journal of Physical Chemistry C*, 121(19), 10552–10561. <https://doi.org/10.1021/acs.jpcc.7b01749>). Other soft materials of high interest, e.g., 2D organic thin films typically show a critical dose on the order of $20-50 e^{-}/\text{\AA}^2$, and MOF, or COF materials, show a critical dose of $100 - 1000 e^{-}/\text{\AA}^2$. At elevated temperatures as in the in-situ experiments, the critical dose is expected to be even lower. Thus, we believe being able to see the structural evolution at the demonstrated spatial resolution for this very challenging soft material is already enough to illustrate the figure of merit of the technique.

Nevertheless, we managed to include an additional material system, a 2D single crystal thin film of α,ω -DH6T bilayer, to more concretely demonstrate the figure of merit of 4D-SCED. Here, we determine the rotation angle using either NBD or SCED at identical sample region and dose conditions. This comprises figure 2 and an additional paragraph, as marked highlight in the revised version. The original figure 2 and 3 are combined and the text adapted to these changes.

6. Why the length of the scale bars in Figure 5f is shorter than the other images? For clear comparisons, the scale bars should be the same.

In the 4D-SCED experiments, the pixel size (sampling size) can be adjusted freely. Together with the number of probed pixels (e.g., 200×200), they define the field of view. During the in-situ observation using 4D-SCED, we started with sampling size of 3.906 nm/pixel and data acquisition over 180×180 pixels. This defines a horizontal field of view of $\sim 700 \text{ nm}$. Approaching the end of the experiment, we noticed the DRCN5T crystal domains have grown quite large. To capture statistically relevant data in larger field of view, we decided (during the experiment) to increase the sampling size to 5.86 nm/pixel and the pixel numbers to 200×200 . When the data are finally visualized, the scale bar has to adapt to the true length scale in the image. We can crop the image and scale it to make the scale bars match the other images, but we prefer to keep the larger field of view to show the almost random orientation of the edge-on domains, as well various size of face-on domains. In the revised version, a note (highlighted) is added to the end of the figure caption.

7. The writing of this manuscript is too colloquial. It is suggested to explain related ideas and results through the third person perspective.

While preparing the manuscript, we followed the suggestion in “how to write your paper” from Nature journals (<https://www.nature.com/nature-portfolio/for-authors/write>) that “Nature journals prefer authors to write in the active voice (“we performed the experiment...”)...”

In the revised version, we have polished the English writing, corrected many typos, grammar mistakes in the initial submitted version, but kept the overall style of active voice.

Reviewers' Comments:

Reviewer #1:

Remarks to the Author:

The authors has addressed my comments satisfactorily and the manuscript is acceptable for publication.

Reviewer #2:

Remarks to the Author:

I appreciate the substantial revisions the authors have made to improve their manuscript. I think the explanations of how their method compares to existing approaches is much improved. I have just one further change that might help the reader. In the discussion on line 197 about the angular resolution being of the order of d/z , it would be helpful to point out again that d depends on α through equation 1. The trade-off between spatial and angular resolution that is present in all diffraction experiments is then clear. Equation one show that increasing α improves the angular resolution through reducing the in-focus spot size, but the equation just below line 192 (which should be labelled equation 2) shows that increasing α worsens the spatial resolution (ΔR becomes larger). A similar trade-off also applies to defocus, z . Larger z gives better angular resolution but worsened spatial resolution. Making these trade-offs clear is important for other who are wishing to select appropriate parameters for their own experiments.

With that minor addition, I am happy for the work to be published.

Reviewer #3:

Remarks to the Author:

The author handled my comments well. The current version can be acceptable without further revisions.

Point-by-point response to comments of reviewer

Reviewer #1 (Remarks to the Author):

The authors have addressed my comments satisfactorily and the manuscript is acceptable for publication.

We sincerely thank the reviewer for his/her efforts to evaluation of our work and constructive comments that have helped to improve our manuscript.

=====

Reviewer #2 (Remarks to the Author):

I appreciate the substantial revisions the authors have made to improve their manuscript. I think the explanations of how their method compares to existing approaches is much improved. I have just one further change that might help the reader. In the discussion on line 197 about the angular resolution being of the order of d/z , it would be helpful to point out again that d depends on α through equation 1. The trade-off between spatial and angular resolution that is present in all diffraction experiments is then clear. Equation one show that increasing α improves the angular resolution through reducing the in-focus spot size, but the equation just below line 192 (which should be labelled equation 2) shows that increasing α worsens the spatial resolution (ΔR becomes larger). A similar trade-off also applies to defocus, z . Larger z gives better angular resolution but worsened spatial resolution. Making these trade-offs clear is important for other who are wishing to select appropriate parameters for their own experiments.

With that minor addition, I am happy for the work to be published.

We appreciate the detailed evaluation of our work.

The suggested statements about trade-offs between spatial and angular resolution is added in the revised manuscript in line 200-209 with highlights.

=====

Reviewer #3 (Remarks to the Author):

The author handled my comments well. The current version can be acceptable without further revisions.

We thank the reviewer for the positive evaluation of our work.